# A Experimental Study on Engineered Cementitious Composites (ECC) Incorporated with *Sporosarcina pasteurii*

**Bingcheng Chen**[1] , **Lufei Du** [1], **Jun Yuan** [2], **Xichen Sun** [3], **Madura Pathirage** [4] , **Weiwei Sun** [1,*] and **Jun Feng** [3,*]

1. Department of Civil Engineering, Nanjing University of Science and Technology, Nanjing 210094, China; chenbingcheng912@163.com (B.C.); dulufei_njust@163.com (L.D.)
2. College of Civil Engineering, Nanjing Forestry University, Nanjing 210037, China; tjyuanjun@163.com
3. National Key Laboratory of Transient Physics, Nanjing University of Science and Technology, Nanjing 210094, China; sunxichen0046@163.com
4. Department of Civil and Environmental Engineering, Northwestern University, Evanston, IL 60208, USA; madura.pathirage@u.northwestern.edu
* Correspondence: sww717@163.com (W.S.); jun.feng@njust.edu.cn (J.F.)

**Abstract:** Microbial-induced calcium carbonate precipitation (MICP) has been successfully applied to self-healing concrete with improved mechanical properties, while the performance of engineered cementitious composites (ECC) incorporated with bacteria is still lacking. In this study, *Sporosarcina pasteurii*, which has a strong ability to produce calcium carbonate, was introduced into engineered cementitious composites (ECC) with mechanical properties analyzed in detail. A multiscale study including compression, tension and fiber pullout tests was carried out to explore the *Sporosarcina pasteurii* incorporation effect on ECC mechanical properties. Compared with the control group, the compressive strength of S.p.-ECC specimens cured for 7 days was increased by almost 10% and the regained strength after self-healing was increased by 7.31%. Meanwhile, the initial crack strength and tensile strength of S.p.-ECC increased by 10.25% and 12.68%, respectively. Interestingly, the crack pattern of ECC was also improved to some extent, e.g., bacteria seemed to minimize crack width. The addition of bacteria failed to increase the ECC tensile strain, which remained at about 4%, in accordance with engineering practice. Finally, matrix/fiber interface properties were altered in S.p.-ECC with lower chemical bond and higher frictional bond strength. The results at the microscopic scale explain well the property improvements of ECC composites based on the fine-scale mechanical theory.

**Keywords:** microbial-induced calcium carbonate precipitation (MICP); engineered cementitious composites (ECC); *Sporosarcina pasteurii*; strength regain; interface alteration

## 1. Introduction

With high compressive strength, good durability and low cost, concrete is the most widely used building material in the world [1–3]. However, due to its disadvantages such as low tensile strength, high brittleness and poor crack resistance, it is susceptible to the influence of the complex surrounding environment during long-term use, making it difficult to avoid micro-cracks on the surface and inside of concrete. The cracks that occur not only affect the strength of the concrete, but also corrosive substances such as water, oxygen and chlorides in the environment can invade the concrete through cracks, causing a series of problems such as leakage, carbonation and steel corrosion, reducing the durability of concrete structures. If these micro-cracks are effectively repaired before the formation of macro-cracks, the safety and durability of the structure will be significantly improved [4].

Microbial-induced calcium carbonate precipitation (MICP) [5,6] is considered to be a method of crack healing. When concrete cracks, the bacteria attached to the surface of the cracks are activated by contacting with air and water, and precipitate calcium carbonate to repair the cracks [7–11]. Among them, *Sporosarcina pasteurii*, with its high activity

and alkali resistance, is used to induce calcium carbonate precipitation [12]. At suitable temperatures (∼30 °C) this bacterium produces urease most effectively, which hydrolyzes urea to ammonia and carbon dioxide. Ammonia production increases the pH value of bacteria in the surrounding environment, thus promoting the precipitation of calcium carbonate in calcium rich environment. The overall chemical reaction is as follows [13,14]:

$$CO(NH_2)_2 + 3H_2O \xrightarrow{\text{Urease}} 2NH_4^+ + 2OH^- + CO_2 \uparrow \tag{1}$$

$$CO_2 + OH^- \rightarrow HCO_3^- \tag{2}$$

$$HCO_3^- \xrightarrow{OH^-} CO_3^{2-} + H^+ \tag{3}$$

$$Cell + Ca^{2+} + CO_3^{2-} \rightarrow Cell + CaCO_3 \downarrow \tag{4}$$

Chahal et al. [15] found that the addition of *S. pasteurii* can improve the compressive strength of concrete, repair concrete cracks and reduce the porosity and permeability of concrete. Due to the harsh environment in concrete, researchers have utlized carrier to protect bacteria and improve their viability in concrete. Encapsulating bacteria with expanded perlite, Mohamed et al. [16] found that cracks with a width of 0.4 mm could be completely healed. Khaliq et al. [17] reported that graphite nanoplates could be used as bacterial immobilization carriers to help concrete heal cracks with a width of 0.6 mm. In addition, the immobilized bacteria such as diatomite, ceramsite and hollow fiber also show very effective self-healing phenomenon in concrete. However, these carriers are not an ideal choice because of their light density which will decrease the strength of concrete. Recently, Bundur et al. [18] proposed that the air-entraining agent (AEA) produces additional pores in the mortar matrix, which provides a certain living space for microorganisms without affecting the concrete strength. Although the healing width is reduced, it can repair the cracks of about 0.3 mm [19–21].

Engineering cement-based composite (ECC) has been extensively studied as a new type of ductile cementitious material with strain hardening characteristics since its appearance in 1990s [22–25]. The difference between ECC and normal concrete is that it can form several small cracks to obtain super-high tensile ductility (>3%) [26]. Figure 1 shows a typical uniaxial tensile stress-strain curve [27–29]. After the first crack, the tensile load capacity continues to increase, resulting in a macroscopic strain-hardening phenomenon accompanied by multiple micro-cracking, and failure after reaching the ultimate strength. Based on the micro fiber bridging and micro mechanical design framework, the average crack width in ECC can be controlled within 60–80 μm [30], which makes it particularly suitable for the development of bacterial based ductile concrete without relying on special carriers. As mentioned above, compared with the immobilized bacteria technology, the bacterial concrete with air entraining agent can heal cracks with a width of 0.3 mm, but it is still far beyond the typical crack width range of ECC material.

Although ECC is capable of filling cracks by further hydration through an incompletely reacted gelling material, it is limited to cracks up to 50 μm [31–34]. It would be highly desirable to fill larger cracks with bacterial mineralization-induced calcium carbonate. On the other hand, it is also worth exploring that the addition of small amounts of bacteria may change the mortar matrix and fiber/matrix interface properties of ECC, thus affecting ECC macroscopic mechanical properties.

The focus of this study is the effect of bacterial incorporation on the mechanical properties of ECC. Firstly, the design theory of ECC materials was briefly introduced. Then the self-healing agent was prepared by sequestering the bacteria with air-entraining agent to provide certain ecological niches for them. Macroscopic mechanical experiments related to ECC, including compression and uniaxial tensile tests, were conducted to analyze whether the incorporation of bacteria poses an effect on the ECC strength and ductility performance. Finally, single fiber pull-out tests were also carried out to explore the effect of *S. pasteurii* on the fiber/matrix interface at the microscopic level.

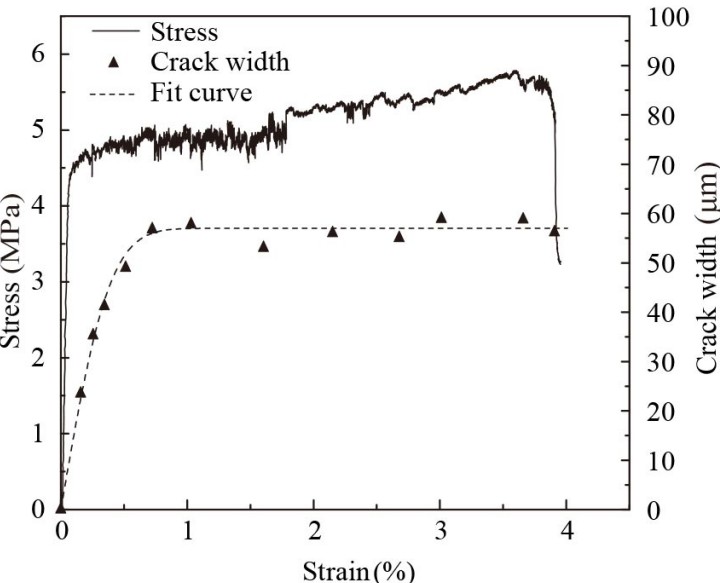

**Figure 1.** A typical uniaxial tensile stress-strain curve of ECC [30].

## 2. Consideration of ECC Design

The formation of multi-cracking and strain-hardening behaviors under tension is a primary task in ECC material design. To achieve such remarkable tensile ductility, the following strength criterion and energy criterion [22,27] must be satisfied:

$$\sigma_0 > \sigma_{fc} \tag{5}$$

$$J_{tip} \leq \sigma_0\delta_0 - \int_0^{\delta_0} \sigma(\delta)d\delta \equiv J_b' \tag{6}$$

where $\sigma_0$ is the maximum crack bridging stress; $\sigma_{fc}$ is the initial crack strength of the mortar matrix; $J_b'$ is the complimentary energy; $J_{tip}$ is the fracture energy of the mortar matrix.

In order to satisfy the strength criterion, sufficient fiber bridging capacity must be ensured at the time of microcrack initiation at the defect. That is, it requires that the initial crack strength $\sigma_{fc}$, which is controlled by the fracture toughness of the matrix and the size of the initial flaw, is less than the fiber bridging capacity $\sigma_0$ at any potential crack surface. The energy criterion is to ensure steady state crack propagation in the case of multi-cracking. This requires that the crack tip toughness $J_{tip}$ should be less than the complimentary energy $J_b'$ calculated from the fiber bridging stress-crack opening ($\sigma$-$\delta$) curve. It can be understood that after subtracting the energy consumed by opening the initial crack from the energy provided by the external load, there should be enough energy for the crack to propagate in a steady state manner. These two criteria must be satisfied to realize the tensile strain hardening behavior, otherwise the tensile softening behavior will occur, whereby the most obvious phenomenon is local fracture.

## 3. Experimental Program

In this section, the cultivation of bacteria, the preparation of bacteria incorporated ECC and its related mechanical tests were carried out to illustrate whether the incorporation of *Sporosarcina pasteurii* has a positive effect on ECC.

### 3.1. Bacterial Strains and Cultures Preparation

In this work, *Sporosarcina pasteurii* from the Beijing biological preservation center (ATCC 11859) was selected to study the self-healing ECC material. *S. pasteurii* not only has superior performance in calcium carbonate production, but also can resist high alkali and survive in harsh environment of mortar matrix [35,36].

The medium for *S. pasteurii* cultivation is consisted of 0.13 mol/L Tris (pH = 9), 20 g/L yeast extract, 10 g/L ammonium sulfate and 5 g/L sodium chloride. After preparing the medium solution, it was put into 121 °C autoclave for sterilization. Then, the activated *S. pasteurii* was inoculated into the sterilized medium in the super clean platform, and cultured for 24 h under condition of 30 °C and vibration of 220 rpm. In order to evaluate the bacteria growth, the concentration of bacteria was measured by spectrophotometer (AQ8100, Shanghai) every two hours. The optical density curve of bacterial growth is shown in Figure 2. *S. pasteurii* grew rapidly in the first 24 h, and then entered a stable phase after 48 h, whereby the cell density was about $1.95 \times 10^{7}$ cells/ml with 2.4 $OD_{600}$ value. In biological studies, $OD_{600}$ indicates the absorbance value at 600 nm, i.e., the higher absorbance value represents higher bacteria concentration. At the same time, the pH of the bacterial solution increased with the growth of the bacteria and eventually stabilized at about 9. This is because bacteria decompose urea to produce a large amount of hydroxide, which leads to an increase in the alkalinity of the solution. This environment has a facilitating effect on the precipitation of calcium carbonate. Figure 3 shows the morphology of bacteria observed under scanning electron microscope (SEM) after centrifugation and freeze-drying of bacterial solution cultured for 48 h, which is long strip, 1.32 μm in length and 0.58 μm in width.

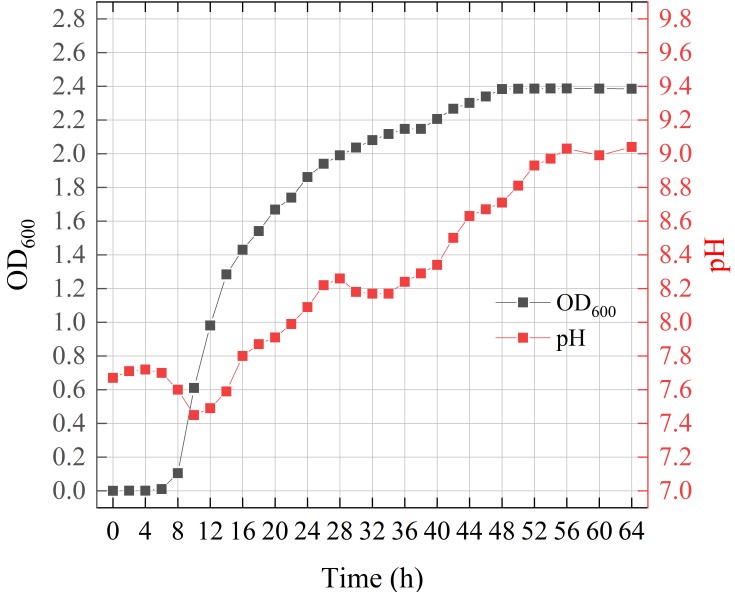

**Figure 2.** Monitoring of growth and pH changes in cultivation process.

A certain amount of sterile water was added to the bacterial solution cultured for 48 h, and the bacterial concentration was diluted to $2 \times 10^{6}$ cells/ml. According to literature reports [37,38], the concentration of bacteria in this state has an obvious effect on the crack healing and strength improvement. Furthermore, 20 g/L calcium chloride was added to the diluted bacterial solution as self-healing agent [39] for subsequent S.p.-ECC preparation.

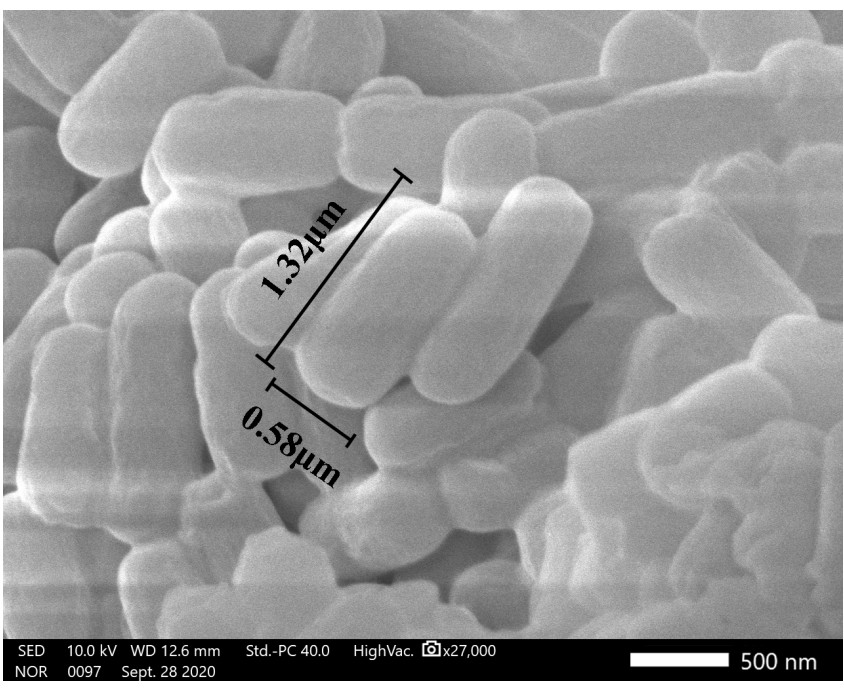

**Figure 3.** *S. pasteurii* morphology under SEM.

### 3.2. Raw Materials and Mixture Proportions

Table 1 lists the detailed mix proportions of ECC used in this study. The main materials for preparing ECC include the following materials: Ordinary portland cement (CEM I 52.5), grade F fly ash (FA), quartz sand, water, superplasticizer (SP), PVA fibers, $CaCO_3$ whisker, air-entraining agent (AEA) and self-healing agent. The water binder ratio and sand binder ratio are set to 0.3 and 0.36, respectively. PVA fibers produced by Kuraray Co. Ltd. of Japan is adopted (2% of mortar volume), and its specific physical and mechanical properties are given in Table 2. $CaCO_3$ whisker is a micro-scale fibrous material with high tensile strength, which can enhance the stability of the ECC mixture. The saponin AEA [18,40] produced by Jinan Chuangli Chemical was used to produce a large number of tiny bubbles of 10–600 μm [41] in diameter, which provide specific ecological niches for bacteria. The SP is polycarboxylic acid water reducing agent produced by Jiangsu subot company to improve the fluidity of fiber-reinforced matrix. Except for the self-healing agent, the mixture proportions of S.p.-ECC and control-ECC are the same.

**Table 1.** Mix proportion of ECC (kg/m$^3$).

| Group | Cement | Fly Ash | Water | Quartz Sand | SP | CaCO$_3$ Whisker | AEA [a] | PVA Fibers [b] | Healing Agent |
|---|---|---|---|---|---|---|---|---|---|
| Control-ECC | 311.8 | 1247.2 | 467.7 | 561.3 | 31.2 | 15.8 | 0.01% | 29.5 | 0 |
| S.p.-ECC | 311.8 | 1247.2 | 467.7 | 561.3 | 31.2 | 15.8 | 0.01% | 29.5 | $2 \times 10^6$ cells/mL |

[a] by weight of binder (cement and fly ash). [b] 2% of the mortar volume.

**Table 2.** Physical and mechanical properties of the PVA fibers.

| Diameter (mm) | Elongation (%) | Density (g/cm$^3$) | Length (mm) | Elastic Modulus (GPa) | Tensile Strength (MPa) |
|---|---|---|---|---|---|
| 0.039 | 7 | 1.3 | 12 | 42.8 | 1690 |

All solid ingredients, including cement, fly ash, quartz sand and $CaCO_3$ whiskers, were first mixed for 3 min. Then the mixing liquid (water, AEA and self-healing agent) and superplasticizer were added to the dry mixture and mixed for 2 min. When the fresh mortar reached a homogeneous state, PVA fibers were slowly added and mixed for 1 min until the fibers were uniformly distributed. Finally, the fresh mixture was poured into the

casting mold and vibrated fully for 1 min. All specimens were demolded after 24 h and stored in a standard curing room at temperature of ∼20 °C and relative humidity ≥ 90%.

### 3.3. Experimental Tests

The effect of *S. pasteurii* incorporation on the mechanical properties of ECC was investigated herein. The compression test and uniaxial tensile test of ECC material were carried out to explore the changes of compressive strength, tensile strength, tensile strain capacity and crack morphology. At the same time, influence of bacteria on the fiber/matrix interface was investigated at the microscale by single fiber pull-out tests.

The uniaxial compression test specimens were 40 × 40 × 40 mm in size. Under condition of a loading rate of 2 mm/min, the compressive strength of 7 days and 28 days are measured, respectively. The compressive strength of each ECC mixture is determined by three valid test data. Meanwhile, uniaxial tensile tests were carried out on specimens cured for 28 days to evaluate the tensile properties of ECC material. The tensile test setup and geometric dimensions of the specimens are shown in Figure 4. Sample sizes in the gauge area are 80 mm (length) × 30 mm (width) × 13 mm (thickness). Due to the relatively small cross section area of the gauge, most cracks are likely to occur in the gauge region during tension, which ensures the reliability of the tensile strain measurement [42]. Two external Linear Variable Differential Transformers (LVDTs) are installed at the end of the gauge area of the specimen to record the displacement for strain calculation. The tensile test was performed under the displacement control of 0.3 mm/min to simulate the quasi-static loading conditions. At the end of the test, the number of cracks and the crack width of the ECC specimens after tension were recorded [43,44].

The mechanical properties of the fiber/matrix interface were characterized by single fiber pullout tests, and the interface parameters such as chemical bond ($G_d$), frictional bond strength ($\tau_0$) and slip-hardening coefficient ($\beta$) were obtained, of which detailed parameter formulas can be found in Refs. [45,46]. The sample preparation is shown in Figure 5, the PVA fiber is fixed equidistantly in the middle of the mold with double-sided tape, and then the fresh mortar matrix (without fiber) is poured into the mold. After 24 h, the sample was demolded for sequent standard curing. Before the fiber pullout tests, the mortar matrix were cut with 1.5 mm thickness for each specimen, ensuring that the debonding process could be captured successfully when fibers were pulled out. Figure 6 shows the detailed fiber pullout tests setup. The sample was fixed on a 50 N precision load cell, the free end of the fiber was fixed on the aluminum plate with strong glue, which was fixed on the top gripper. The test was displacement controlled with loading speed of 0.3 mm/min [43].

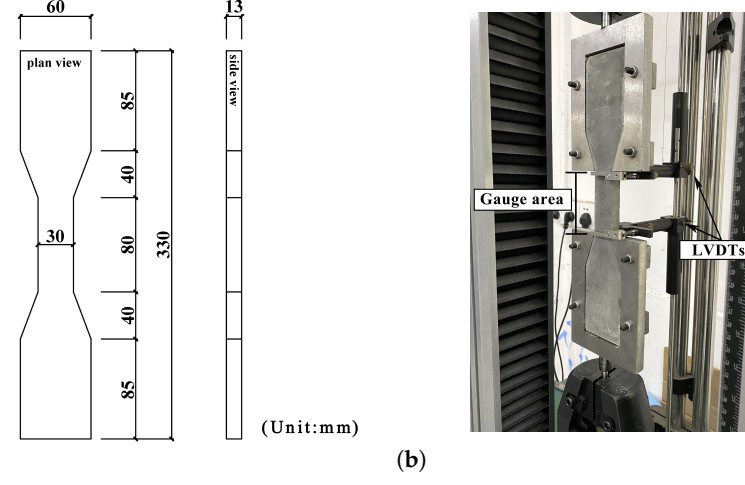

**Figure 4.** Uniaxial tensile test. (**a**) Geometry of the specimen and (**b**) schematic diagram of the test apparatus.

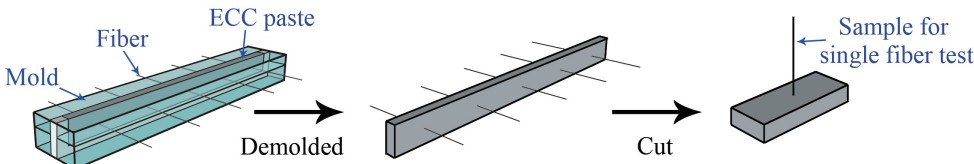

**Figure 5.** Schematic diagram of single fiber/matrix sample preparation.

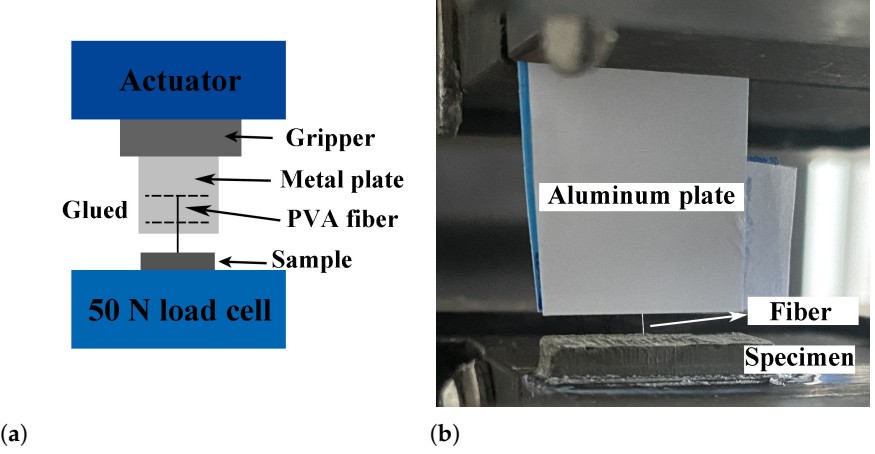

(**a**)　　　　　　　　　　　　　　　　　　　　(**b**)

**Figure 6.** Single fiber pull-out tests. (**a**) Detailed description of test and (**b**) schematic diagram of test.

## 4. Experimental Results

### 4.1. Compressive Strength

Two groups of ECC mixtures were subjected to compression tests, and the compressive strength at 7 and 28 days of curing was measured. In order to evaluate the strength regain by bacteria, all samples were compressed after the peak load with same residual strength and then healed for 28 days, the curing method has been described in detail in the previous study [19,38]. Then the compression tests were conducted to observe the change of compressive strength before and after self-healing. The relevant data have been listed in Tables 3 and 4 (C7-1 represents the first specimen in the control group that has been cured for 7 days). The failure pattern of ECC and S.p.-ECC compressed specimens after complete damage is shown in Figure 7.

Figure 8 shows the compressive strength of ECC mixture cured at 7 days. In Figure 8a, it can be seen that the compressive strength of both control-ECC and S.p.-ECC increased rapidly in the first 7 days. The average compressive strength of the two groups were 38.39 MPa and 39.65 MPa, indicating that the incorporation of bacteria did not adversely affect the compressive strength of the ECC mixture. After compression to the same residual strength, the specimens were healed and then subjected to compression again. Compared with the residual strength, the strength of the two groups of ECC mixture was improved, and the average recovery strength was 0.72 MPa and 2.98 MPa, respectively. Compared with the control group, the compressive strength of the S.p.-ECC mixture was recovered by 7.31%, as shown in Figure 8b. On one hand, the recovery of compressive strength is due to the unhydrated cementitious material continue to participate in the reaction, on the other hand, it is because the calcium carbonate induced by bacteria fills the pores in the concrete, which changes the microstructure of ECC, reduces the porosity and improves the compressive strength.

At the same time, the compression test of ECC mixture cured for 28 days was also carried out. As shown in Figure 9, compared with the specimens cured for 7 days, the compressive strength of the control-ECC and S.p.-ECC was 42.97 MPa and 46.82 MPa, respectively. After healing, uniaxial compression test was carried out again, and the strength of the two groups did not increase compared with the residual strength, because the dominant hydration reaction basically ended. However, comparing the recovery strength of two groups of ECC mixture, it can be found that the average recovery strength of S.p.-ECC is 12.2% higher than that of the control group, indicating that calcium carbonate produced by bacteria has a positive effect on the improvement of compressive strength which corresponds to previous studies [15,47].

**Table 3.** Compressive strength of the two ECC mixtures recovered after early curing (7 days). (unit: MPa).

| Group | Compressive Strength (7 Days) | Residual Strength | Healing Strength | Recovered Strength |
|-------|-------------------------------|-------------------|------------------|--------------------|
| C7-1 | 37.94 | 31.31 | 32.13 | 0.81 |
| C7-2 | 38.86 | 31.31 | 31.33 | 0.29 |
| C7-3 | 38.36 | 31.33 | 32.39 | 1.06 |
| S7-1 | 40.03 | 31.18 | 33.16 | 1.98 |
| S7-2 | 39.88 | 31.29 | 35.08 | 3.79 |
| S7-3 | 39.03 | 32.23 | 35.39 | 3.17 |

**Table 4.** Compressive strength of the two ECC mixtures recovered after standard curing (28 days). (unit: MPa).

| Group | Compressive Strength (28 Days) | Residual Strength | Healing Strength | Recovered Strength |
|-------|--------------------------------|-------------------|------------------|--------------------|
| C28-1 | 42.91 | 31.79 | 25.74 | −6.05 |
| C28-2 | 43.38 | 31.74 | 24.23 | −7.51 |
| C28-3 | 42.61 | 31.51 | 20.15 | −11.36 |
| S28-1 | 46.28 | 31.34 | 26.64 | −4.70 |
| S28-2 | 47.04 | 31.32 | 24.81 | −6.51 |
| S28-3 | 47.13 | 31.63 | 27.21 | −4.43 |

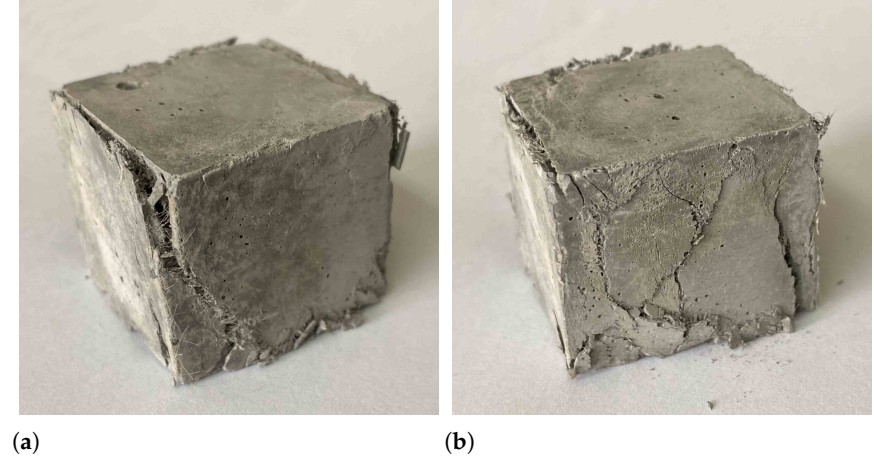

(**a**)          (**b**)

**Figure 7.** Comparison of final damage patterns of uniaxial compression specimens. (**a**) Control-ECC. (**b**) S.p.-ECC.

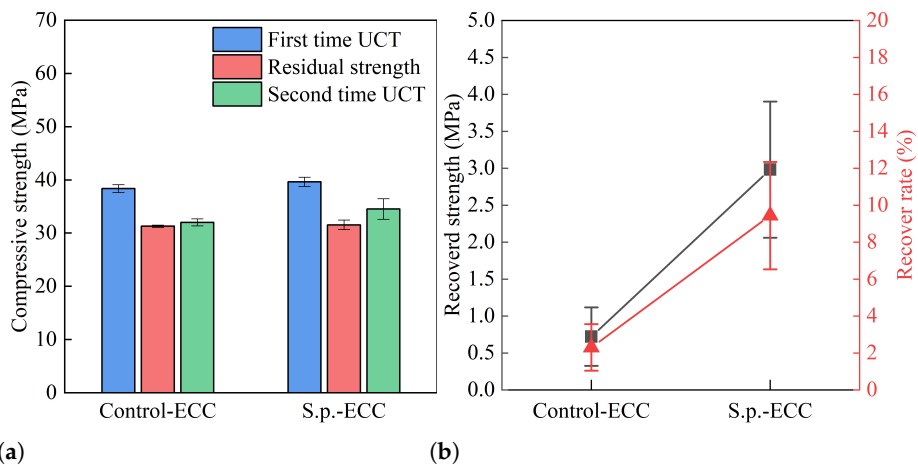

**Figure 8.** Strength recovery of ECC samples at early age (7 days). (**a**) Compressive strength for repeated compression tests. (**b**) Strength recover rate.

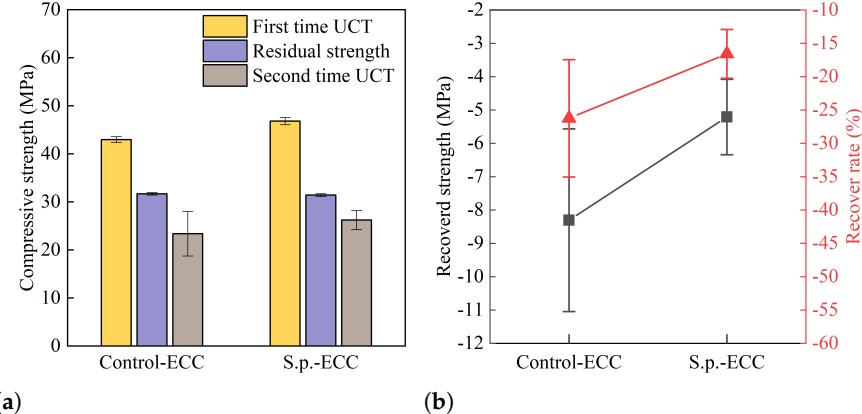

**Figure 9.** Strength recovery of ECC samples at Standard cured (28 days). (**a**) Compressive strength for repeated compression tests. (**b**) Strength recover rate.

### 4.2. Uniaxial Tensile Behaviors

The effect of *S. pasteurii* on the tensile properties of the composites was tested by the uniaxial tensile test of dumbbell-shaped specimens, as shown in the tensile stress-strain curve in Figure 10. From these curves, the relationship among elastic modulus, first crack strength, tensile strength and tensile strain capacity of ECC mixture can be obtained. The crack development at different strain stages is also analyzed, point A shows the generation of the main crack when the ECC specimen reaches the first crack strength, which almost runs through the specimen section along the direction of stress. Points B, C and D are the crack development corresponding to each 1% strain interval, and it can be seen that when the main crack is produced, the specimen is not further damaged here, but many tiny cracks are produced around it. Point E shows the final crack distribution, when the strain reaches the maximum and finally damage occurs along the main crack and the stress drops. Figure 11 shows the experimental curves of the two kinds of specimens, and the relevant attributes are summarized, as shown in Table 5.

It can be seen from Figure 11a,b that compared with the control group, the incorporation of *S. pasteurii* has little effect on the tensile strain capacity of ECC material, and both show apparent tensile strain hardening behavior, with the average strain capacity of 4.07% and 4.08%, respectively. As shown in Figure 12, the tensile strength of ECC mixed with bacteria was 5.51 MPa, which was 12.7% higher than that of the control group (4.89 MPa). At the same time, the matrix properties of the ECC mixture also changed, and the elastic modulus

increased from 16.39 GPa to 17.52 GPa. The tensile test results show that the incorporation of bacteria has a positive regulatory effect on the mechanical properties of ECC mixture.

Unlike conventional brittle concrete, ECC has unique ability to control the width of cracks [48,49]. Therefore, the crack width is also considered a significant material property, which affects the structure's durability. The typical cracking pattern of the specimen after the tensile test is shown in Figure 13, it can be roughly seen that the cracks of the S.p.-ECC are finer and denser. The optical microscope was used to observe and record the cracks, with an accuracy of 10 μm, and the statistical results are shown in Figure 14. Compared with the control-ECC, the maximum crack width of S.p.-ECC decreased while the narrow cracks increased. This observation seems to suggest that the S.p.-ECC specimen may have lower water permeability and aggressive ion permeability coefficient, leading to better durability.

Table 6 shows the average crack width and the total number of cracks for ECC mixtures. It can be seen that S.p.-ECC causes more cracks while the average fracture width decreases. As mentioned above, crack width has an important effect on self-healing of concrete. Due to the addition of self-healing agents to ECC, more cracks appeared.

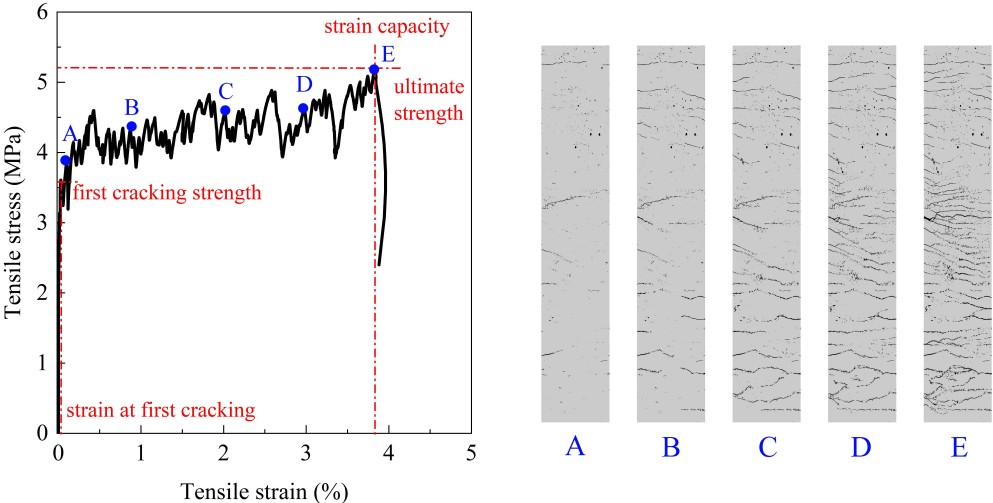

**Figure 10.** Stress-strain curves and crack development of ECC mixture.

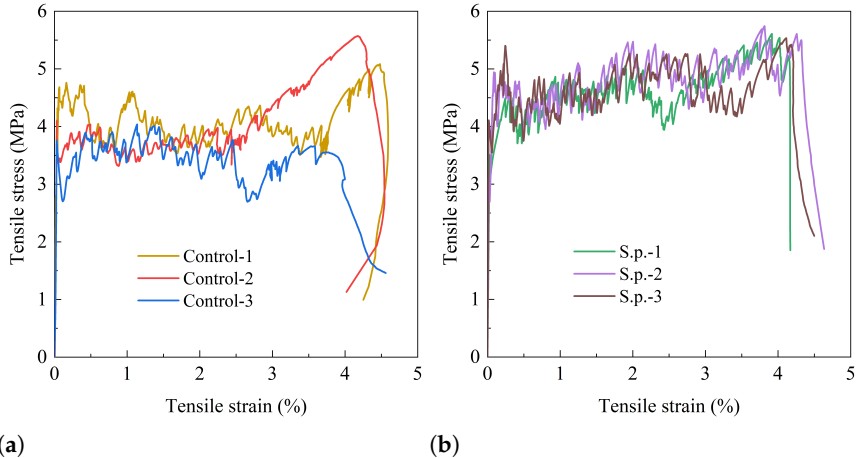

(**a**)　　　　　　　　　　　　　　　　　　　　　　(**b**)

**Figure 11.** Tensile stress-strain experimental curve of ECC. (**a**) Without bacteria. (**b**) With bacteria.

**Table 5.** Summarized tensile properties of ECC mixtures. (unit: MPa).

| Group | Elastic Modulus (GPa) | First Crack Strength | Ultimate Tensile Strength | Strain Capacity (%) |
|-------|----------------------|---------------------|--------------------------|---------------------|
| C-1 | 16.49 | 3.02 | 5.07 | 4.48 |
| C-2 | 14.89 | 3.51 | 5.57 | 4.18 |
| C-3 | 17.79 | 3.14 | 4.03 | 3.55 |
| S-1 | 21.21 | 3.76 | 5.19 | 3.84 |
| S-2 | 15.34 | 4.10 | 5.60 | 4.15 |
| S-3 | 16.01 | 2.80 | 5.74 | 4.25 |

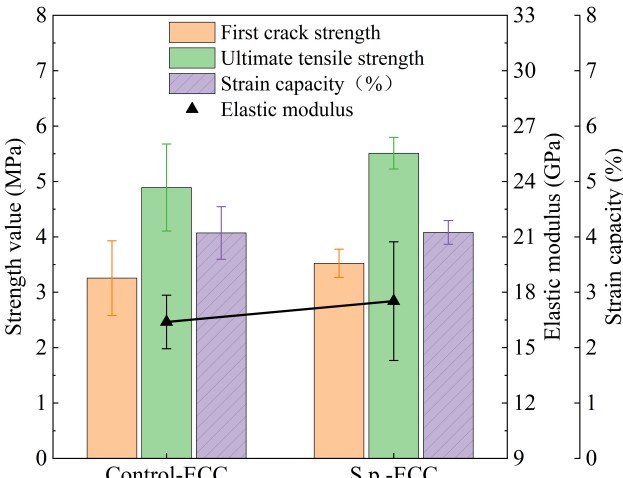

**Figure 12.** Comparison of the properties of ECC tensile specimens.

**Table 6.** Crack statistics of ECC tensile specimens.

| Mixture | Control-ECC | S.p.-ECC |
|---------|-------------|----------|
| Average crack width | 63 μm | 56 μm |
| Crack number | 38 | 42 |

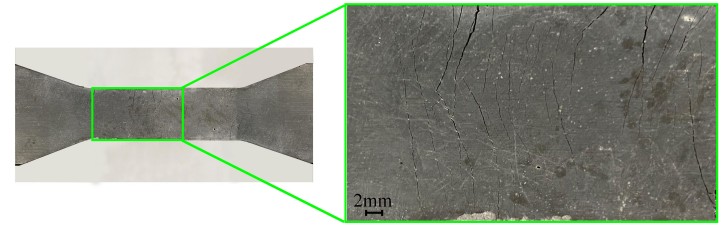

(**a**)

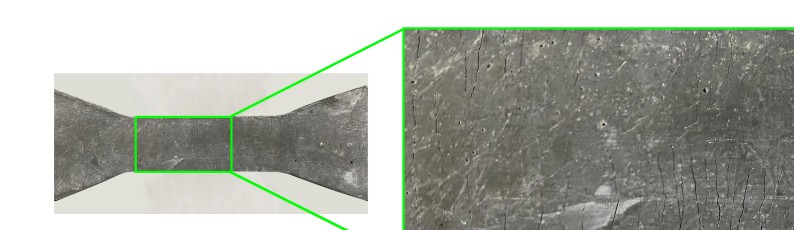

(**b**)

**Figure 13.** Cracking pattern of (**a**) control-ECC (**b**) S.p.-ECC.

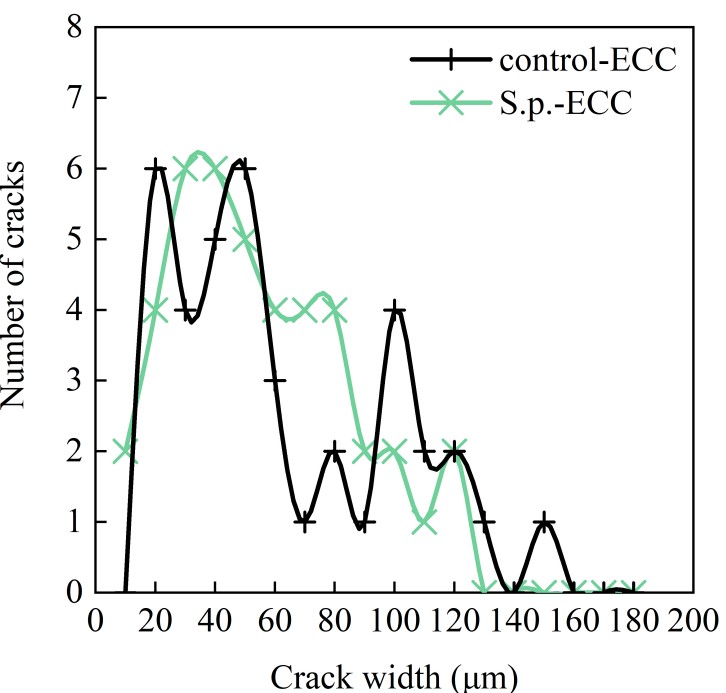

**Figure 14.** Crack width distribution in ECCs.

### 4.3. Single Fiber Pullout Behavior

Fiber pullout includes two stages, i.e., debonding stage and slippage stage, as shown in Figure 15. In the debonding stage, the pullout force was mainly caused by the bonding of the fiber segment in the debonding zone to the interface. The longer the debonding distance, the greater the pulling force. The sudden load drop corresponds to the full debonding of fiber from the matrix, which divides the fiber pullout process into the preceding debonding stage and the following slippage stage. Two constitutive parameters, namely chemical bond ($G_d$) and friction bond ($\tau_0$), can be deduced from the single fiber pullout curve to describe the interfacial bond properties using following equations.

$$G_d = \frac{2(P_a - P_b)^2}{\pi^2 E_f d_f^3} \tag{7}$$

$$\tau_0 = \frac{P_b}{\pi d_f L_e} \tag{8}$$

where $P_a$ is the peak pullout force, $P_b$ is the sudden drop load, $E_f$ is the elastic modulus of the fiber, $d_f$ is the fiber diameter and $L_e$ is the fiber embedment length. The curve's curvature during the pullout process reflects the change of interfacial friction bond, which is represented by the slip hardening parameter $\beta$ [46,50].

The single fiber pullout curves of the two ECC mixtures are shown in Figure 16, and at least five valid data are obtained for each group. At the same time, the fiber/matrix interface parameters, including $G_d$, $\tau_0$ and $\beta$, were calculated for each group of curves, as shown in Figure 17 and Table 7. Compared with control-ECC, the chemical bond of S.p.-ECC decreased from 0.885 J/m$^2$ to 0.231 J/m$^2$. The lower chemical bond means that the fibers are more likely to detach from the surrounding matrix and enter the slip stage, which is essential for continued cracking of the matrix. The decrease of chemical bond may be due to certain bacterial mineralization products attached to the surface of the PVA fiber. To test this hypothesis, a portion of the PVA fiber was taken from the fracture of the ECC specimen after stretching and observed with SEM. As shown in Figure 18, many bacterial mineralization products were attached to the surface of the fibers in S.p.-ECC group, whereas this was not the case for the fibers of the control-ECC. In addition to the

original coating of the fibers, the encapsulation of precipitated material on the PVA fibers further reduces the $Al^{3+}$ and $Ca^{2+}$ concentrations at the fiber/matrix interface. Since these two ions are essential for the formation of a strong interfacial thin layer between the PVA polymer and the surrounding hydration products, their lower concentrations will reduce the chemical bond.

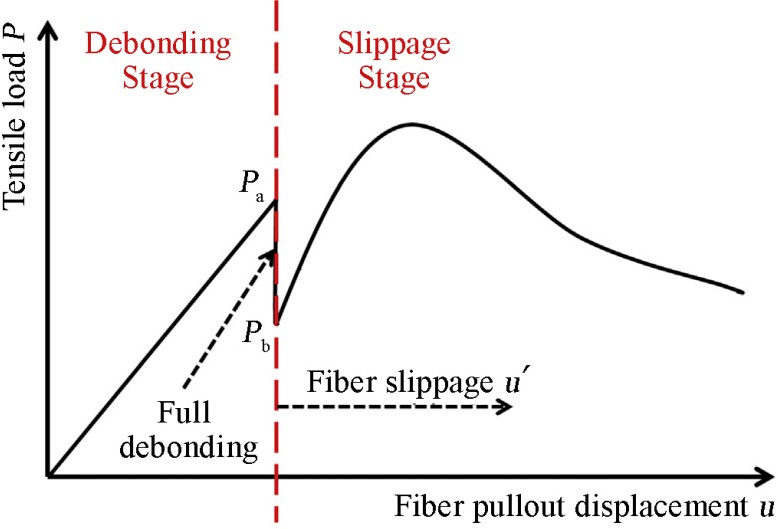

**Figure 15.** Illustration of typical single-fiber pullout force-displacement curve.

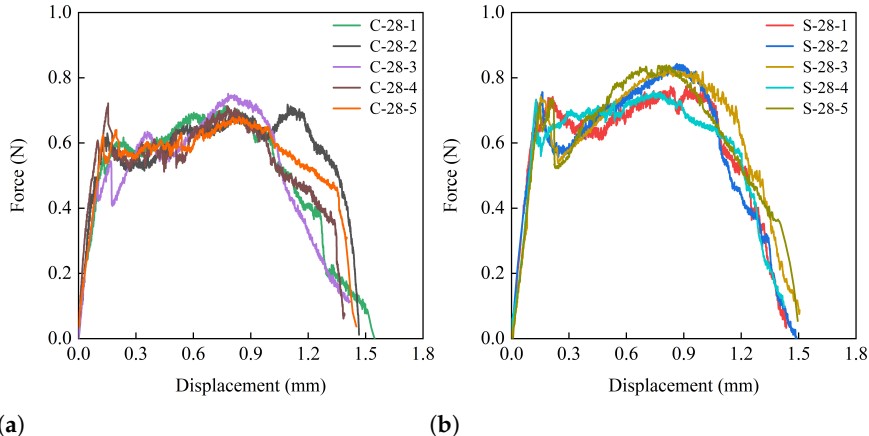

(**a**)                                                                                      (**b**)

**Figure 16.** Single fiber pullout curve (**a**) control-ECC (**b**) S.p.-ECC.

The difference is that friction bonding strength ($\tau_0$) and slip hardening coefficient $\beta$ increase with the addition of bacteria. The average friction bonding strength ($\tau_0$) showed a 49.6% improvement, and the slip hardening coefficient $\beta$ increase from a negligible value of 0.15 to 0.31. The mineralization of bacteria may reduce the lubricating effect of oil coating on fiber surface, which leads to the rise of friction bonding strength and slip hardening coefficient. The higher friction bonding strength, the greater the interface friction to resist fiber slippage. As a result, the width of cracks decreases and the number of cracks increases, which can be clearly observed in S.p.-ECC samples, as shown in Figure 13.

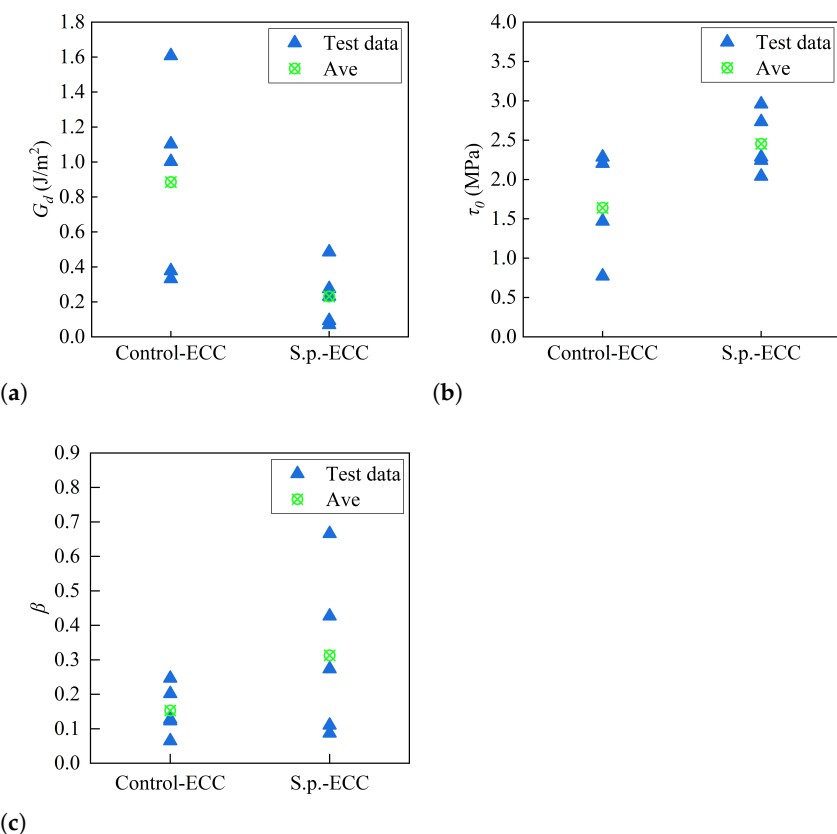

(**a**)

(**b**)

(**c**)

**Figure 17.** Comparison of interface parameters for single fiber pullout tests. (**a**) Chemical bond. (**b**) Frictional bond strength. (**c**) Slip-hardening coefficient.

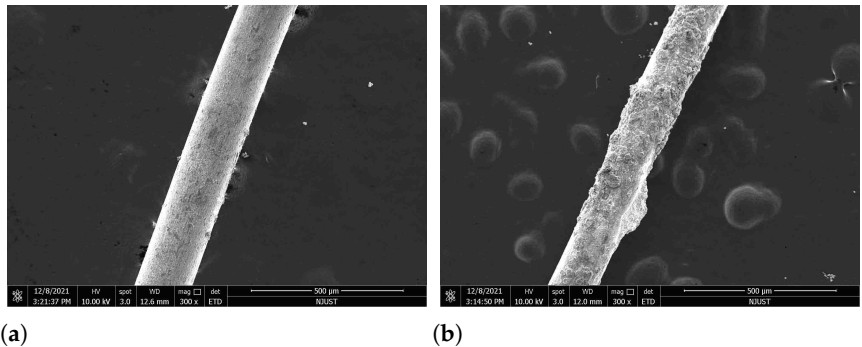

(**a**)

(**b**)

**Figure 18.** PVA surface comparison under SEM observation. (**a**) control-ECC. (**b**) S.p.-ECC.

**Table 7.** Comparison of mean values of interface parameters for single fiber pullout test.

|  | Chemical Bond (J/m$^2$) | Frictional Bond Strength (MPa) | Slip-Hardening Coefficient |
|---|---|---|---|
| Control-ECC | 0.885 ± 0.535 | 1.640 ± 0.621 | 0.15 ± 0.07 |
| S.p.-ECC | 0.231 ± 0.167 | 2.453 ± 0.380 | 0.31 ± 0.24 |

To summarise, simple bacterial treatment significantly reduces the chemical bond, because it potentially provides an alternative way for fiber surface treatment and tailoring high-performance ductile cement-based composites. In practice, the reduction of chemical bond was realized by the relatively expensive oiling process during traditional PVA ECC manufacturing [50].

## 5. Conclusions

This paper focuses on the multiscale study of mechanical properties of engineered cementitious composites incorporated with *Sporosarcina pasteurii*. Uniaxial compression and uniaxial tensile tests were conducted to investigate the bacteria effects on compressive strength, strength recover, tensile strength, tensile strain capacity and crack morphology. The fiber/matrix interface parameters were also compared via single fiber pull-out tests. The following conclusions were drawn:

(1) The incorporation of *Sporosarcina pasteurii* had a positive effect on the compressive strength and strength recovery of ECC specimens. The compressive strength of S.p.-ECC specimens cured for 7 days was increased by almost 10% and the regained strength after self-healing was increased by more than 7%.

(2) The incorporation of bacteria failed to increase the tensile strain of ECC specimens, which could reach an average strain of 4.08%. At the same time, the matrix properties of ECC material changed with a decrease in crack width and an increasing number of narrow cracks, indicating better durability. Meanwhile, the initial crack strength and tensile strength of S.p.-ECC increased by 10.25% and 12.68%, respectively. In conclusion, the incorporation of S. pasteurii led to a general improvement of the tensile properties of ECC.

(3) At the microscopic level, calcium carbonate precipitates produced by bacteria wrapped the fiber surface and changed the interfacial properties. The *Sporosarcina pasteurii* incorporation decreases the chemical bond from 0.885 J/m$^2$ to 0.231 J/m$^2$, while increasing the average friction bonding strength ($\tau_0$) by 49.6%, and improves the slip hardening coefficient $\beta$, increasing from 0.15 to 0.31.

**Author Contributions:** B.C.: Methodology, visualization, writing—original draft: L.D.; investigation: J.Y.; conceptualization: X.S.; investigation: M.P.; writing—review and editing: W.S.; supervision, funding acquisition: J.F.; methodology, writing—review and editing. All authors have read and agreed to the published version of the manuscript.

**Funding:** This research was funded by the National Natural Science Foundation of China, grant number 11902161.

**Institutional Review Board Statement:** Not applicable

**Informed Consent Statement:** Not applicable

**Data Availability Statement:** The data that support the findings of this study are available on request from the corresponding author.

**Acknowledgments:** This research was financially supported by the National Natural Science Foundation of China (No. 11902161) and the Major Science and Technology Projects in Shanxi under Grant 20201102003. Weiwei Sun thanks the Foundation Strengthening Plan Technology Fund (No. 2019-JCJQ-JJ-371). Bingcheng Chen expresses gratitude to the Postgraduate Research & Practice Innovation Program of Jiangsu Province (No. KYCX20_0377). Lufei Du thanks the Postgraduate Research & Practice Innovation Program of Jiangsu Province (No. SJCX22_0129).

**Conflicts of Interest:** The authors declare that they have no known competitive financial interests or personal relationships that may affect the work reported herein.

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
