# Peer review of "A Experimental Study on Engineered Cementitious Composites (ECC) Incorporated with Sporosarcina pasteurii"

_buildings, doi:10.3390/buildings12050691_

Round 1

Reviewer 1 Report

The article includes a multiscale study including compression, tension and fiber pullout tests to explore the Sporosarcina pasteurii incorporation effect on ECC mechanical properties. The experimental activity was well-conceived and performed correctly. Nevertheless, some issues are not correctly addressed, and the article needs revisions before being suitable for publication. Some detailed comments are provided below:

  • Abstract:
    • The gap needs to be stated in the abstract of the paper.
  • Introduction:
    • The gap in the paper is not clear. The authors should state what is done and what is not in the literature as well as the novelty of the publication.
  • Materials and methods:
    • The authors should state the reason of the chosen self-healing procedure employed (using the residual strength after the peak compressive strength).
  • Experimental results:
    • I would remove Table 3 and Table 4 (or include them in the supplementary material) because they possess the same information than Figure 8 and Figure 9.
    • I would remove Figure 12 (or include them in the supplementary material) because they possess the same information than Table 5.
    • There is no or little discussion of the mechanisms behind the effect of adding Sporosarcina pasteurii on the ECC. The paper just lists the results of the different experiments with no further discussion. This should be improved to be considered for publication.
  • Conclusions:
    • The conclusions of the paper are an overview of the results instead of a summarize discussion of the findings of the paper. This should be improved.

Reviewer 2 Report

The study examines the Sporosarcina pasteurii bacteria incorporation effect on engineered cementitious composites (ECC) in terms of mechanical and fiber/matrix interface properties. This is a good experimental work that can be improved with minor revision. The paper is in line with the scope of the Journal. I suggest reviewing the following:

  1. In the Abstract, change the misspelled “examed” with “examined”. Besides, in the same sentence, how conclusions about the chemical bond can be drawn from a physical test such as single fiber pullout test? I suggest rephrasing this sentence.
  2. For the statement in lines 75-76 please add reference.
  3. Figure 3, please check the scale. Either the bar at the bottom or the measurements are wrong.
  4. Please specify which is the binder (i.e., cement + fly ash) in the notation of Table 1.
  5. In the Raw materials and mixture proportion section, if possible specify the superplasticizer and air-entraining agent used in the mixes. Furthermore, is there any reason mixes where vibrated for so long? (3 minutes seems a lot to me, which increases the risk of segregation)
  6. Lines 121-122 please specify at what age were carried out the tensile tests.
  7. Figure 4, add unit.
  8. Line 155, please check the % or rephrase explaining better. Where that 8.93% comes from?
  9. The results of recovered strength (both at 7 and 28 days) have a high dispersion, and this makes the conclusions less sound. Further investigation would be needed to prove that S. pasteurii are effectively contributing to the strength recovery after peak load is reached. Additionally, the recovery of compressive strength for samples tested at 7 days is due to the further hydration on binder that at early age is not completed (i.e., unhydrated binder). Authors stated this in Lines 159-162. However, is difficult to know the contribution of each process (e.g., further hydration or calcium carbonate precipitation).
  10. Check punctuation in Lines 177-180.
  11. Lines 188-190, I suggest to soften this statement. The error for the Control-ECC is very high. There is almost no improvement on the tensile strength (as the author also mentioned in other parts of the manuscript).
  12. Lines 202-203, I would remove the sentence. This conclusion is not very clear from the results of this study, besides, if there are references it means that it has already been observed.
  13. Please describe how the chemical bond (Gd) results were calculated.
  14. Lines 235-236, please clarify.
  15. In the Conclusions section, number (1), how can authors state that the porosity was reduced if no test to assess porosity was performed? It is a hypothesis… consider removing from the conclusions.
  16. Lines 100 and 101 add “s” to fiber, and Line 120 remove “s” to strengths.

Reviewer 3 Report

The paper “A multiscale study on engineered cementitious composites (ECC) incorporated with Sporosarcina pasteurii” presents an interesting study. To improve the quality of the paper, please consider the following comments on the paper before resubmission.

  1. Please remove the multiscale word from the title of the paper.
  2. Page-2, line 10, When cracks occur, not only will they affect the strength of concrete, but also corrosive substances such as water, please revise the statements considering English, and make it in present form.
  3. Page-2, please mention the temperatures required for the reaction of equations (1, 2,3,4) to happen.
  4. Page-3, Engineering cement-based composite (ECC) has been extensively studied as a new type of ultra-ductile cementitious material, please revise it as concrete/mortar cannot be classified as ultra-ductile material.
  5. In the conclusion, the results of the tests are presented in percentages, please use the approximate word before the values in percentage and round it. For instance, instead of 7.31%, you could write more than 7%.

Round 2

Reviewer 1 Report

I would recommend this paper for publication